# Transcriptome-wide identification and expression profiling of the ERF gene family suggest roles as transcriptional activators and repressors of fruit ripening in durian

**Gholamreza Khaksar**[1], **Supaart Sirikantaramas**[1,2]*

**1** Molecular Crop Research Unit, Department of Biochemistry, Faculty of Science, Chulalongkorn University, Bangkok, Thailand, **2** Omics Sciences and Bioinformatics Center, Chulalongkorn University, Bangkok, Thailand

* supaart.s@chula.ac.th

**Data Availability Statement:** The data that support the findings of this study are openly available in NCBI Sequence Read Archive (SRA) under the project accession number [PRJNA683229] and

## Abstract

The involvement of the phytohormone ethylene as the main trigger of climacteric fruit ripening is well documented. However, our knowledge regarding the role of ethylene response factor (ERF) transcription factor in the transcriptional regulation of ethylene biosynthesis during fruit ripening remains limited. Here, comprehensive transcriptome analysis and expression profiling revealed 63 *ERF*s in durian pulps, termed *DzERF1–DzERF63*, of which 34 exhibited ripening-associated expression patterns at three stages (unripe, midripe, and ripe) during fruit ripening. Hierarchical clustering analysis classified 34 ripening-associated *DzERF*s into three distinct clades, among which, clade I consisted of downregulated *DzERF*s and clade III included those upregulated during ripening. Phylogenetic analysis predicted the functions of some DzERFs based on orthologs of previously characterized ERFs. Among downregulated *DzERF*s, DzERF6 functional prediction revealed its role as a negative regulator of ripening via ethylene biosynthetic gene repression, whereas among upregulated genes, DzERF9 was predicted to positively regulate ethylene biosynthesis. Correlation network analysis of 34 ripening-associated DzERFs with potential target genes revealed a strong negative correlation between DzERF6 and ethylene biosynthetic genes and a strong positive correlation between DzERF9 and ethylene biosynthesis. *DzERF6* and *DzERF9* showed differential expression patterns in association with different ripening treatments (natural, ethylene-induced, and 1-methylcyclopropene-delayed ripening). *DzERF6* was downregulated, whereas *DzERF9* was upregulated, during ripening and after ethylene treatment. The auxin-repressed and auxin-induced expression of *DzERF6* and *DzERF9*, respectively, confirmed its dose-dependent responsiveness to exogenous auxin. We suggest ethylene- and auxin-mediated roles of DzERF6 and DzERF9 during fruit ripening, possibly through transcriptional regulation of ethylene biosynthetic genes.

CNGB Sequence Archive (CNSA) of China National
GeneBank DataBase (CNGBdb) with project
accession number [CNP0001432].

**Funding:** This research was funded by the National
Research Council of Thailand (NRCT5-RSA63001-
15) and Chulalongkorn University (GRU
6203023003-1) (to S.S.).

**Competing interests:** The authors have declared
that no competing interests exist.

## Introduction

Fruit ripening is a complex developmental and genetically programmed process that leads to marked changes in the texture, flavor, color, and nutritional value of the flesh. These changes occur as a result of the coordinated activation of numerous biochemical and genetic pathways regulated by transcriptional and hormonal regulatory networks [1]. Fruit ripening is categorized as climacteric and non-climacteric ripening based on ethylene production and the respiration rate over the course of ripening [2]. Climacteric fruit ripening is associated with a burst in ethylene biosynthesis and respiration at the onset of ripening. The gaseous phytohormone ethylene not only plays a key regulatory role in climacteric fruit ripening but is also involved in various developmental and physiological processes, including programmed cell death, flowering, seed germination, and responses to both biotic and abiotic stressors [3, 4].

Ethylene response factors (ERFs) belong to the large superfamily of APETALA2/ethylene response factor (AP2/ERF) transcription factors (TFs), which function at the end of the ethylene signaling pathway [5, 6]. ERF TFs mediate ethylene-responsive gene expression by specifically binding to the GCC box (AGCCGCC) and/or dehydration-responsive element/C-repeat (DRE/CRT) (CCGAC) located in the promoter regions of numerous ethylene-regulated genes. The defining characteristic of the members of the ERF TF family is the conserved DNA binding domain (AP2/ERF domain), which consists of approximately 60 amino acids and binds to the promoters of target genes [7]. The AP2/ERF superfamily is further divided into four families, ABI3/VP (RAV), AP2, ERF, and soloist, according to the number of AP2/ERF domains. The ERF family of TFs harbors a single AP2/ERF domain. Members of the RAV family have an AP2/ERF and a B3 domain, whereas AP2 family TFs usually contain multiple repeated AP2/ERF domains. Soloist family members show low similarity to other members [6, 8].

An increasing number of studies has identified the members of the AP2/ERF superfamily in various crops, such as tomato [9], rice [10], kiwifruit [11], barley [12], apple [13], plum [14], longan [15], and Chinese jujube [16]. The roles of ERFs in a wide range of plant physiological and developmental processes, including fruit ripening, have been well documented [17–19]. As the core of ethylene signaling, the roles of ERFs in regulating different aspects of climacteric fruit ripening have been extensively studied, including color change (pear [20]; tomato [21]), fruit softening (tomato [22]; kiwifruit [23]; banana [24]; papaya [25]), and aroma formation (banana [26]).

A huge body of evidence suggests that the regulation of climacteric fruit ripening depends mainly on the modulation of ethylene biosynthesis and/or signaling. The autoregulation of ethylene biosynthesis via the transcriptional regulation of ethylene biosynthetic genes (1-aminocyclopropane-1-carboxylic acid (ACC) synthase; *ACS* and ACC oxidase; *ACO*) is a consequence of the ethylene response in ripening fruits [23, 27]. Hence, the identification and functional characterization of ERFs would provide a deeper understanding of ethylene-related ripening regulation. However, few studies have addressed the possible role of ERFs in the transcriptional regulation of ethylene biosynthetic genes in relation to fruit ripening. Lee et al. [21] suggested that tomato ERF (SlERF6) is a transcriptional repressor of ripening because the downregulation of *SlERF6* results in higher expression levels of ethylene biosynthetic genes (ACC synthase; *ACS2* and ACC oxidase; *ACO1*) and increased ethylene biosynthesis. In banana, MaERF11 suppresses the expression of *MaACS1* and *MaACO1* [28], whereas MaERF9 was reported to activate the expression of *MaACO1*, suggesting its role as a transcriptional activator of banana fruit ripening [24]. In apples, MdERF2 acts as a transcriptional repressor of ripening by suppressing the expression of *MdACS1* [29].

Durian (*Durio zibethinus* L.) is an economic tropical fruit crop that belongs to the family Malvaceae and is native to Southeast Asia. Durian has gained an ever-increasing popularity

among consumers both locally and in the international market because of its unique and overwhelming flavor, described as having a sweet taste with a sulfuryl and sweet fruity odor. With more than 200 cultivars, Thailand is the top exporter of durian across the Southeast Asian region. However, a few cultivars are commercially cultivated and in high demand, including Monthong (*D. zibethinus* Murr. cv. 'Monthong') and Chanee (*D. zibethinus* Murr. cv. 'Chanee'). Among these, Monthong is of great interest owing to its creamy texture and mild odor [30]. Durian is a climacteric fruit with a short shelf life. The ultimate goal is to offer durian fruit with a longer shelf life, which has remained a challenge for the agricultural industry. To achieve this, gaining a deeper understanding of the molecular mechanisms underlying the ripening process is essential.

The draft genome of durian was previously released [31], which enabled further studies on the identification of TFs regulating fruit ripening on a genome-wide scale. Previously, we conducted a genome-wide analysis of the Dof (DNA binding with one finger) TF family and identified 24 durian *Dof*s (*DzDof*s), of which 15 were expressed in the fruit pulp. The functional characterization of *DzDof2.2* suggested a role during fruit ripening by regulating auxin biosynthesis and auxin–ethylene crosstalk [32]. In another study, we identified a member of the auxin response factor (ARF) TF family, DzARF2A, which mediates durian fruit ripening through the transcriptional regulation of ethylene biosynthetic genes [33]. Using metabolome and transcriptome analyses, Sangpong et al. [34] investigated dynamic changes in the contents of flavor-related metabolites during the post-harvest ripening of durian fruit and identified key genes involved in their biosynthetic pathways. These reports provide us with a better understanding of the transcriptional and hormonal regulatory networks involved in durian fruit ripening. However, knowledge of ERF TFs in durian fruit and their possible roles in regulating post-harvest ripening is still lacking. Herein, to address this, we conducted a transcriptome-wide analysis and identified 34 ripening-associated DzERFs. We then profiled their expression levels with exogenous ethylene and auxin treatments. Our findings provide insights into the role of ERF TFs in mediating the post-harvest ripening of durian fruit and lay a foundation for further investigations of the ethylene regulatory network in durian fruit ripening.

## Materials and methods

### Plant materials and treatments

Durian (*Durio zibethinus* L.) fruit, cv. Monthong, was harvested from a commercial durian orchard located in the Trat province in the eastern part of Thailand. Fruit samples of similar size and weight (~3–4 kg each) were collected at the commercially mature stage, which was 105 days after anthesis. Three types of samples (unripe, midripe, and ripe) were used in our study. Fruits harvested at the mature stage were used as unripe fruit samples. To obtain midripe and ripe fruit samples, fruits harvested at the mature stage were kept at room temperature (30˚C) for post-harvest ripening until reaching a firmness of 3.4 ± 0.81 N (~3 days after harvest) (for midripe stage) and 1.55 ± 0.45 N (~5 days after harvest) (for ripe stage) [32, 34] and then were peeled. After peeling the fruit samples, two central pulps were collected and processed following the method described by Pinsorn et al. [30]. A texture analyzer was used to measure the firmness of the first pulp as the indicator of fruit ripening [32]. Thereafter, the second fruit pulp was collected, immediately frozen in liquid nitrogen, and stored at −80˚C until RNA extraction. Pictures of representative durian fruit pulps at unripe, midripe, and ripe stages are presented in S1 Fig.

To profile the expression levels of candidate ripening-associated *DzERF*s under ethylene treatment, three different ripening conditions were used, natural, ethephon-induced, and 1-methylcyclopropene (1-MCP)-delayed ripening. Fruit samples of the Monthong cultivar

were harvested at the mature stage and treated with either ethephon (48% 2-chloroethylphosphonic acid; Alpha Agro Tech Co., Ltd., Thailand; for ethephon-induced) or 1-MCP (0.19% 1-MCP tablet; BioLene Co., Ltd., China; for 1-MCP-delayed ripening). Briefly, the ethephon solution (69.35 mg/mL) was exogenously applied to the upper area of each fruit stalk. Regarding the 1-MCP treatment, each fruit sample was put inside a closed 20-L chamber. After that, one tablet of 1-MCP was placed into a beaker inside the chamber. Water (5 mL) was added to the beaker which then generated gaseous 1-MCP (19.54 ppm) and the chamber was immediately closed for 12 h at room temperature (30 ˚C). As control, samples were kept under similar conditions without 1-MCP [32]. Thereafter, the control and treated samples were kept at room temperature (30˚C) (for ∼3 days) until the ethephon-induced samples ripened. Then, all samples were peeled, and the collected pulps were stored at −80˚C for further analysis (RNA extraction).

For exogenous auxin application, young leaves of durian tree cv. Monthong were soaked in different concentrations (10, 20, and 40 µM) of indole-3-acetic acid (IAA) (Duchefa Biochemie, The Netherlands) for 2 h. For control, leaves were soaked in distilled water without IAA. After treatment, leaves were immediately frozen in liquid nitrogen and stored at -80˚C until total RNA extraction [33].

## Transcriptome analysis

To obtain transcriptome data for durian fruit cv. Monthong at the three stages of post-harvest ripening (unripe, midripe, and ripe), sequencing reads from the RNA-Seq study of durian fruit cv. Monthong (generated by our group) were retrieved from a public repository database with the following accession number: PRJNA683229 [34].

## Mapping the reads to the *D. zibethinus* reference genome and expression analysis

We used the OmicsBox program (Biobam, Spain) for transcriptome data analysis. Raw reads were filtered to obtain high-quality clean reads by removing adapters, reads shorter than 60 bp, and low-quality reads with a Q-value ≤ 30 using FastQC and Trimmomatic. Then, a gene-level analysis was performed by aligning the reads against the reference genome of durian cv. Musang King [31] using STAR (Spliced Transcripts Alignment to a Reference). Counting of reads and expression analysis were performed with HTSeq-count using default parameters. Transcripts with normalized reads <1 reads per kilobase of exon per million fragments (RPKM) were considered not expressed.

## Transcriptome-wide identification and expression profiling of durian *ERF*s

Based on gene annotation and bioinformatics analysis, we identified 63 genes harboring the AP2/ERF domain as *ERF* family genes in durian (*DzERF*s), termed *DzERF1–DzERF63*. First, we downloaded the hidden Markov model (HMM) file corresponding to the AP2/ERF domain (PF00847) from the Pfam protein family database (http://pfam.xfam.org/). We then used it as a query to search against the transcriptome database of durian fruit cv. Monthong, using HMMER software (http://hmmer.org/) with the parameters of score (bits) >200 and e-value cut-off ≤1e$^{-5}$. The amino acid sequences of DzERFs were further confirmed in the SMART database for the presence of the conserved AP2/ERF domain. Hierarchical clustering of the heatmap was performed using MetaboAnalyst 4.0, an open source R-based program [35], to profile the expression levels of *DzERF*s at three different stages (unripe, midripe, and ripe) during the post-harvest ripening of durian fruit cv. Monthong based on transcriptome sequencing

data. The transcripts were represented as the mean of the RPKM value at each ripening stage. This approach enabled us to identify 34 candidate ripening-associated *DzERFs*.

## Multiple sequence alignment and phylogenetic analysis of ripening-associated DzERFs

Multiple sequence alignment of 34 ripening-associated DzERFs was performed using ClustalW (MEGA X software) with default parameters. We constructed a neighbor-joining phylogenetic tree with 1000 bootstrap replicates (with a JTT model and pairwise gap deletion) using MEGA X software by aligning the full-length protein sequences of ripening-associated DzERFs with ERFs from tomato (SlERFs), a model for climacteric fruit, and banana (MaERFs), along with previously characterized ERFs from different climacteric fruit crops, including apple ERF2 (MdERF2 (XP_008376369.1) [29]), pear ERF24 (PpERF24 (XP_009370001.1) [36]), pear ERF96 (Pp12ERF96 (XP_009375859.1) [20]), kiwi ERF9 (AdERF9 (ADJ67438.1) [23]), papaya ERF9 (CpERF9 (XP_021902277.1) [25]), peach ERF2 (PpeERF2 (XP_007210477.2) [37]), and persimmon ERFs (DkERF8/16/9 (AFH56415.1/AID51421.1/AGA15800.1) [38]).

## Conserved motif analysis

We used the MEME program (http://meme-suite.org) to identify the conserved motifs of ripening-associated DzERFs [39] with the following parameter settings: motif length = 6–100; motif sites = 2–120; maximum number of motifs = 10; the distribution of a single motif was "any number of repetitions".

## *In silico* promoter analysis

The 2000-bp upstream promoter regions of ethylene biosynthetic genes of durian [ACC synthase (*ACS*; XM_022901720.1) and ACC oxidase (*ACO*; XM_022903266.1)] were scanned for the ethylene response elements [GCC box (AGCCGCC) and/or dehydration-responsive element/C-repeat [DRE/CRT] (CCGAC)) using the online tool PLACE (http://www.dna.affrc.go.jp/) [40].

## Tissue-specific expression profiling of ripening-associated *DzERFs*

Illumina sequencing reads from the RNA-Seq analysis of durian cv. Musang King were retrieved from a public repository database (SRA, Sequence Read Archive) as follows: SRX3188225 (root tissue), SRX3188222 (stem tissue), SRX3188226 (leaf tissue), and SRX3188223 (aril/pulp tissue). The de novo assembled transcriptome was generated following the method described in our previous study [32]. Then, we aligned the input reads to the assembled transcriptome using the abundance estimation tool in the Trinity package to obtain the raw counts of each contig. A trimmed mean of M-values normalized matrix was generated by merging the raw read counts into a single read count matrix [32]. A heat map was generated to visualize the data using the normalized total read counts as input queries in MetaboAnalyst 4.0. For heatmap construction, the values were sum-normalized, $\log_2$ transformed, and autoscaled.

## Gene network visualization

To investigate and visualize the gene expression correlations of 34 ripening-associated DzERFs with some ripening-related genes previously identified by Teh et al. [31] [from an RNA-Seq study of durian cv. Musang King, including *ACS*, *ACO*, methionine gamma lyase (*MGL*), pectinesterase (*PME40*), S-adenosylmethionine (*SAM*) *synthase*, β-D-xylosidase 1 (*BXL1*), cytochrome P450 71B34 (*CYP71B34*), sulfur deficiency-induced 1 (*SDI1*), and *DPNPH*] and Khaksar et al. [32] [L-tryptophan aminotransferase 1 (*TAA1*) and indole-3-pyruvate

monooxygenase (*YUCCA4*)], the conserved domain of each enzyme (based on an HMM) was first obtained from the Pfam protein database (http://pfam.xfam.org/). This sequence was used as a query to search against the de novo assembled transcriptome database of durian fruit cv. Monthong and the Musang King genome (i.e., *ACS* (XM_022901720.1), *ACO* (XM_022903266.1), *TAA1* (XM_022878297.1), *YUCCA4* (XM_022900772.1), *MGL* (XM_022917834.1), *PME40* (XM_022875865.1), *SAM synthase* (XM_022915017.1), *BXL1* (XM_022866549.1), *CYP71B34* (XM_022919875.1), *SDI1* (XM_022914153)). The network of TFs and candidate target genes was visualized using Cytoscape (v3.7.1, USA). A correlation heatmap was generated using MetaboAnalyst 4.0.

### Reverse transcription quantitative polymerase chain reaction (RT-qPCR)

Total RNA was isolated from fruit pulp samples using PureLink Plant RNA Reagent (Thermo Fisher Scientific™) following the manufacturer's instructions. The extracted RNA was treated with DNase I (Thermo Fisher Scientific) to remove the genomic DNA. To assess the quality and quantity of RNA samples, agarose gel electrophoresis and an Eppendorf BioPhotometer D30 were used with A260/280 and A260/230 ratios from 1.8 to 2.0 and 2.0 to 2.2, respectively, following the standard guidelines described by Bustin et al. [41]. First-strand cDNA synthesis was performed using 1 μg of total RNA using a RevertAid First Strand cDNA Synthesis Kit (Thermo Fisher Scientific™), following the manufacturer's recommended protocol and the standard guidelines of reverse transcription, as described by Bustin et al. [41]. All primers used in this study were designed using Primer 3 online (http://primer3.ut.ee/), which are presented in S1 Table. To measure the transcript levels of candidate ripening-associated *DzERF*s during post-harvest ripening and under exogenous ethylene and auxin treatments, RT-qPCR was performed using a CFX95 Real-time System (Bio-Rad Laboratories Inc., California, USA). The PCR reaction was carried out in a total volume of 10 μL, with 1 μL of diluted cDNA (1 ng of cDNA), 5 μL of 2× QPCR Green Master Mix LRox (biotechrabbit, Berlin, Germany), and 200 nM of each gene-specific primer. The following conditions were used: initial activation at 95°C for 3 min, followed by 40 cycles of denaturation at 95°C for 15 s, annealing at 58–60°C for 30 s, and extension at 72°C for 20 s. Three independent biological replicates were used for each RT-qPCR experiment. The elongation factor 1 alpha (*EF-1α*; XM_022889169.1) and actin (*ACT*; XM_022897008.1) genes of durian were selected as references for the normalization of RT-qPCR data [33]. The normalization of RT-qPCR data was performed using normalization factors, which were the geometric mean of the two reference genes, as stated by Vandesompele et al. [42]. Expression data are presented as fold-change ($2^{-\Delta\Delta Ct}$) [43].

Statistical analysis of gene expression data (RT-qPCR data) was carried out following the method proposed by Steibel et al. [44] which consists of the analysis of cycles to threshold values (Ct) for the target and reference genes according to a linear mixed model (Model I of the report of Steibel et al. [44]). In this model, fixed factors included treatments, genes, and their interactions whereas the obtained Ct values comprised the dependent variables of the model. Samples were the random effects of the model. Statistical significance was set at $P < 0.05$. R programming language (https://www.R-project.org/) was implemented for our analysis.

## Results

### Transcriptome-wide identification and profiling of *ERF*s revealed 34 *DzERF*s with ripening-associated expression patterns

Comprehensive transcriptome identification revealed 63 genes containing the AP2/ERF domain as *ERF* family genes in the pulps of durian (*DzERF*s) (Table 1). The deduced amino acid

**Table 1.  Members of the ERF transcription factor family identified in durian (*Durio zibethinus* L.) pulps.**

| Gene name | Corresponding protein ID | Locus number | Protein length (aa) | MW (kDa) | *p*I value |
|---|---|---|---|---|---|
| *DzERF1* | XP_022757939.1 | LOC111305040 | 320 | 36.00 | 5.68 |
| *DzERF2* | XP_022770870.1 | LOC111314102 | 314 | 35.29 | 5.03 |
| *DzERF3* | XP_022764136.1 | LOC111309326 | 235 | 26.04 | 6.52 |
| *DzERF4* | XP_022739915.1 | LOC111292005 | 212 | 24.62 | 6.05 |
| *DzERF5* | XP_022746214.1 | LOC111296272 | 355 | 38.96 | 6.84 |
| *DzERF6* | XP_022718559.1 | LOC111276884 | 329 | 36.97 | 5.36 |
| *DzERF7* | XP_022728139.1 | LOC111283787 | 138 | 15.59 | 9.25 |
| *DzERF8* | XP_022730575.1 | LOC111285398 | 236 | 25.99 | 8.75 |
| *DzERF9* | XP_022762533.1 | LOC111308473 | 297 | 32.75 | 5.13 |
| *DzERF10* | XP_022738311.1 | LOC111291032 | 237 | 26.28 | 5.89 |
| *DzERF11* | XP_022770063.1 | LOC111313662 | 272 | 30.60 | 6.24 |
| *DzERF12* | XP_022730839.1 | LOC111285586 | 185 | 21.05 | 9.01 |
| *DzERF13* | XP_022769862.1 | LOC111313446 | 138 | 15.77 | 9.74 |
| *DzERF14* | XP_022772517.1 | LOC111315212 | 242 | 27.05 | 5.23 |
| *DzERF15* | XP_022732401.1 | LOC111286611 | 245 | 27.59 | 5.35 |
| *DzERF16* | XP_022750428.1 | LOC111299480 | 266 | 29.07 | 9.20 |
| *DzERF17* | XP_022759480.1 | LOC111305881 | 256 | 28.35 | 5.39 |
| *DzERF18* | XP_022760851.1 | LOC111307063 | 424 | 46.71 | 8.36 |
| *DzERF19* | XP_022717473.1 | LOC111276031 | 143 | 16.03 | 9.75 |
| *DzERF20* | XP_022776484.1 | LOC111318096 | 272 | 30.56 | 8.86 |
| *DzERF21* | XP_022737645.1 | LOC111290559 | 467 | 51.92 | 6.06 |
| *DzERF22* | XP_022759479.1 | LOC111305880 | 271 | 30.51 | 5.32 |
| *DzERF23* | XP_022758322.1 | LOC111305240 | 262 | 28.80 | 8.68 |
| *DzERF24* | XP_022737087.1 | LOC111289975 | 217 | 24.48 | 5.14 |
| *DzERF25* | XP_022735991.1 | LOC111289315 | 324 | 36.37 | 7.74 |
| *DzERF26* | XP_022740848.1 | LOC111292638 | 391 | 43.50 | 5.68 |
| *DzERF27* | XP_022774941.1 | LOC111316955 | 371 | 41.41 | 5.90 |
| *DzERF28* | XP_022742247.1 | LOC111293640 | 357 | 39.96 | 5.95 |
| *DzERF29* | XP_022764049.1 | LOC111309281 | 468 | 51.93 | 5.27 |
| *DzERF30* | XP_022727163.1 | LOC111283042 | 473 | 52.66 | 5.71 |
| *DzERF31* | XP_022774942.1 | LOC111316955 | 472 | 52.59 | 5.83 |
| *DzERF32* | XP_022759482.1 | LOC111305884 | 212 | 23.04 | 9.51 |
| *DzERF33* | XP_022724020.1 | LOC111280803 | 242 | 26.33 | 9.15 |
| *DzERF34* | XP_022736982.1 | LOC111289891 | 277 | 31.19 | 5.32 |
| *DzERF35* | XP_022746038.1 | LOC111296178 | 264 | 29.08 | 4.96 |
| *DzERF36* | XP_022748917.1 | LOC111298470 | 278 | 30.22 | 4.65 |
| *DzERF37* | XP_022740127.1 | LOC111292151 | 166 | 18.59 | 9.73 |
| *DzERF38* | XP_022720025.1 | LOC111277862 | 218 | 24.80 | 4.77 |
| *DzERF39* | XP_022768320.1 | LOC111312376 | 359 | 39.87 | 7.08 |
| *DzERF40* | XP_022735631.1 | LOC111289009 | 428 | 46.88 | 6.07 |
| *DzERF41* | XP_022770537.1 | LOC111313923 | 312 | 35.69 | 5.41 |
| *DzERF42* | XP_022773380.1 | LOC111315706 | 340 | 38.02 | 5.15 |
| *DzERF43* | XP_022773158.1 | LOC111315585 | 361 | 40.42 | 6.08 |
| *DzERF44* | XP_022737593.1 | LOC111290515 | 357 | 39.72 | 5.12 |
| *DzERF45* | XP_022764222.1 | LOC111309418 | 357 | 39.94 | 4.79 |
| *DzERF46* | XP_022723280.1 | LOC111280307 | 392 | 43.50 | 4.76 |
| *DzERF47* | XP_022760599.1 | LOC111306880 | 265 | 29.99 | 6.92 |

(*Continued*)

**Table 1.** (Continued)

| Gene name | Corresponding protein ID | Locus number | Protein length (aa) | MW (kDa) | pI value |
|-----------|--------------------------|--------------|---------------------|----------|----------|
| DzERF48 | XP_022734251.1 | LOC111287840 | 357 | 39.91 | 4.88 |
| DzERF49 | XP_022715360.1 | LOC111274738 | 391 | 43.80 | 4.79 |
| DzERF50 | XP_022722523.1 | LOC111279759 | 262 | 29.79 | 9.43 |
| DzERF51 | XP_022715359.1 | LOC111274738 | 397 | 44.46 | 4.77 |
| DzERF52 | XP_022721652.1 | LOC111279035 | 385 | 43.19 | 5.17 |
| DzERF53 | XP_022749044.1 | LOC111298582 | 261 | 29.35 | 9.05 |
| DzERF54 | XP_022737057.1 | LOC111289954 | 261 | 29.22 | 9.28 |
| DzERF55 | XP_022776509.1 | LOC111318112 | 221 | 25.14 | 6.25 |
| DzERF56 | XP_022723886.1 | LOC111280723 | 205 | 22.80 | 9.47 |
| DzERF57 | XP_022738570.1 | LOC111291204 | 265 | 29.71 | 8.31 |
| DzERF58 | XP_022743982.1 | LOC111294945 | 403 | 44.48 | 5.17 |
| DzERF59 | XP_022742858.1 | LOC111294006 | 272 | 29.80 | 5.41 |
| DzERF60 | XP_022764593.1 | LOC111309859 | 278 | 30.64 | 5.01 |
| DzERF61 | XP_022774944.1 | LOC111316955 | 310 | 34.84 | 6.02 |
| DzERF62 | XP_022743986.1 | LOC111294948 | 225 | 25.65 | 5.23 |
| DzERF63 | XP_022724273.1 | LOC111280980 | 169 | 18.56 | 9.62 |

sequences of DzERFs varied substantially in size, from DzERF7 and DzERF13 with 138 amino acids each to DzERF30 harboring 473 amino acids. Their molecular weights and isoelectric points ranged from 15.59 kDa (DzERF7) to 52.66 kDa (DzERF30) and from 4.65 (DzERF36) to 9.75 (DzERF19), respectively (Table 1). Expression profiling of 63 *DzERF*s at three different stages during post-harvest ripening (unripe, midripe, and ripe) of durian pulp cv. Monthong revealed 34 *DzERF*s with differential expression patterns. A heat map was constructed to cluster these *DzERF*s according to their expression levels (RPKM) based on the Euclidian distance method (Fig 1). Accordingly, the heat map classified 34 *ERF*s into three separate clades. Clade 1 consisted of 15 ERF genes (*DzERF5*, *6*, *11*, *13*, *14*, *18*, *19*, *21*, *23*, *26*, *27*, *28*, *30*, *31*, and *32*), which displayed a decreasing expression pattern during ripening, peaked at the unripe stage, and then declined and reached the lowest expression levels at the ripe stage. Clade 2 included only three members of the ERF gene family (*DzERF16*, *20*, and *22*). The expression levels of these *DzERF*s peaked at the midripe stage and declined at the ripe stage. Clade 3 was made up of 16 genes (*DzERF1*, *2*, *3*, *4*, *7*, *8*, *9*, *10*, *12*, *15*, *17*, *24*, *25*, *29*, *33*, and *34*) strongly upregulated during post-harvest ripening. The expression levels of these *DzERF*s peaked at the ripe stage.

## Multiple sequence alignment and conserved motif analysis

Multiple alignment of the full-length deduced proteins of ripening-associated DzERFs revealed a conserved DNA binding domain of 61 amino acid residues at the N-terminal region, designated the AP2/ERF domain, which is the signature characteristic of members of the AP2/ERF superfamily (S2 Fig). We further investigated the conserved motifs of DzERFs and identified at least 10 (Fig 2 and S3 Fig). Among them, motifs 1 and 2 corresponded to the AP2/ERF domain and were widely distributed in all DzERFs, except for DzERF19, which lacked motif 2 (Fig 2). Notably, the DzERFs that were clustered together harbored a similar motif organization.

## Phylogeny

We constructed a phylogenetic tree of the 34 ripening-associated DzERFs, banana (MaERFs), and tomato (SlERFs) based on the deduced protein sequences. Our phylogenetic analysis

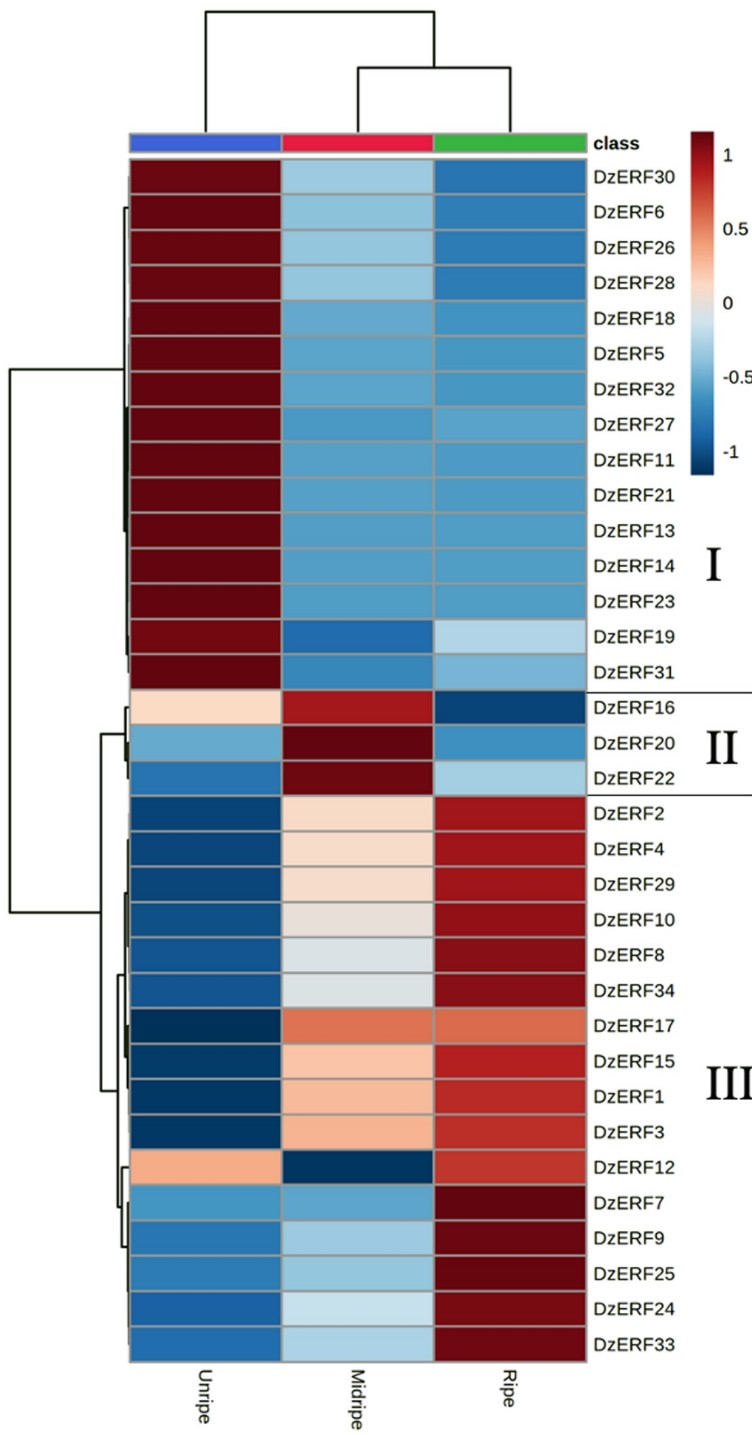

**Fig 1. Heat map of the expression levels of ripening-associated durian ERF family genes at three different stages (unripe, midripe, and ripe) during fruit post-harvest ripening.** The cluster was generated using the Euclidian distance method according to gene expression values (RPKM) from the transcriptome data. For each row, blue and red correspond, respectively, to low and high expression values. For a given gene and stage, the expression value corresponds to the mean of the RPKM value.

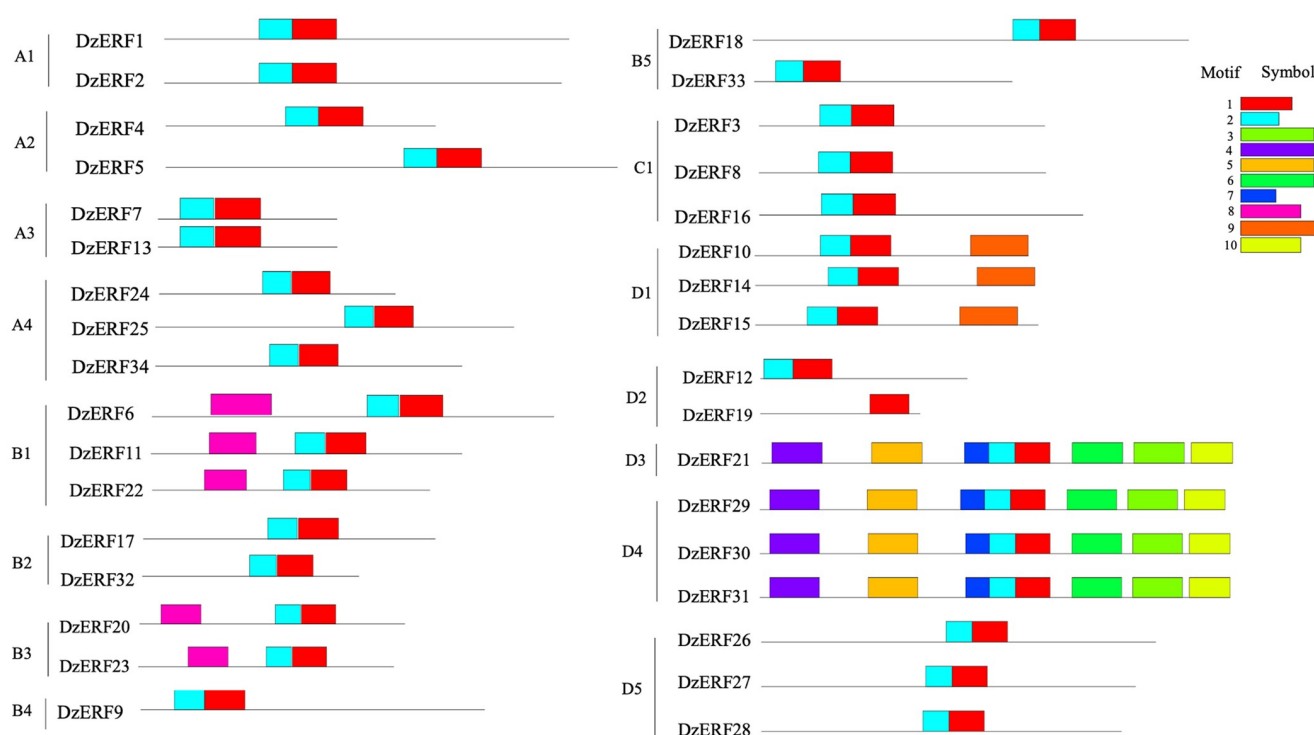

**Fig 2. Motif organization of ripening-associated durian ERFs (DzERFs).** A schematic distribution of 10 conserved motifs identified by MEME suite version 5.1.0. is presented. Motifs 1 and 2 correspond to the DNA binding domain (AP2/ERF domain). The functions of other eight motifs are still unknown and must be further investigated.

revealed that DzERFs were clustered into four clades (A, B, C, and D) and 15 subclades (A1, A2, A3, A4, B1, B2, B3, B4, B5, C1, D1, D2, D3, D4, and D5). Group A4 harbored seven members as the biggest subclade, whereas subclades A2 and D2 were the smallest and each was comprised of only two members (Fig 3).

## Tissue-specific expression of *DzERFs*

From the analysis of publicly available transcriptomic data on different tissues (stem, root, leaf, and pulp) from durian cv. Musang King, we profiled the expression levels of ripening-associated *DzERFs* in different tissues. Notably, three *DzERFs*, including *DzERF9*, *DzERF15*, and *DzERF17* were fruit-specific and were not expressed in other tissues, whereas other *DzERFs* were expressed in all tissues, except for *DzERF24*, which was not expressed in leaf and stem tissues (Fig 4). This expression profile suggests the role of ERFs in a wide range of physiological processes in various tissues.

## Regulatory effects of ripening-associated DzERFs on some target ripening-associated genes

Gene expression correlations of 34 ripening-associated DzERFs with some previously identified ripening-related genes in durian fruit (*SDI1* and *DPNPH*, sulfur metabolism; *SAM synthase*, *ACS*, and *ACO*, ethylene biosynthesis; *MGL*, aroma formation; *PME40* and *BXL1*, cell wall modification; *CYP71B34*, fruit ripening; *TAA1* and *YUCCA4*, auxin biosynthesis) were investigated and visualized as a clustered heatmap (Fig 5A) and a correlation network

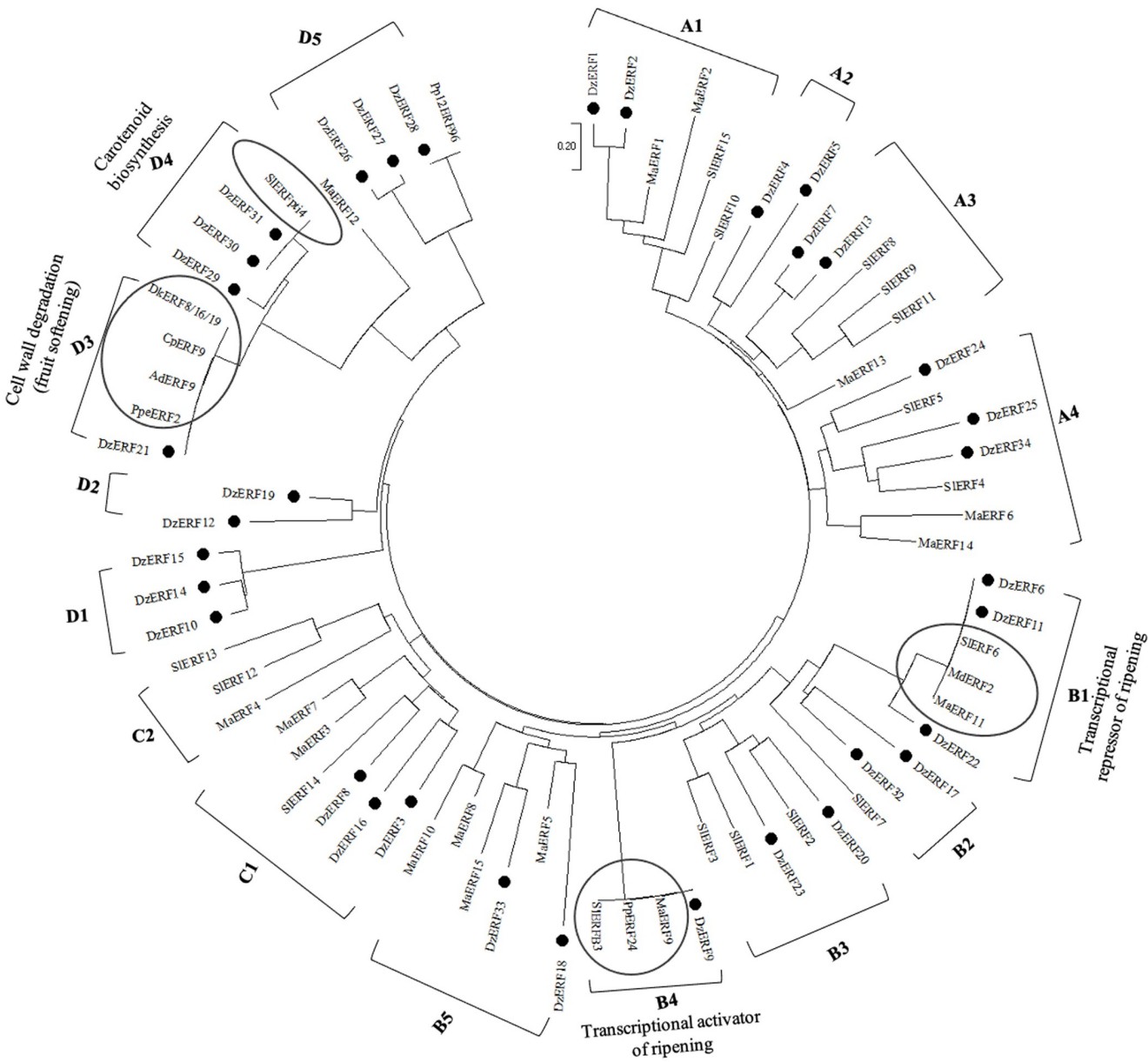

**Fig 3. Phylogenetic tree of the amino acid sequences of the ripening-associated durian ERFs (DzERFs).** The deduced full-length amino acid sequences of DzERFs were aligned with protein sequences of ERFs from tomato (*Solanum lycopersicum*; SlERFs), banana (*Musa acuminata*; MaERFs), and previously characterized ERFs from climacteric fruit crops (apple: MdERFs; pear: PpERFs; papaya: CpERF; kiwi: AdERF; peach: PpeERF; persimmon: DkERFs) to construct the phylogenetic tree using MEGA X software and the neighbor-joining method (with 1000 bootstrap replicates, a JTT model, and pairwise gap deletion using a bootstrap test of phylogeny with the minimum evolution test and default parameters). The previously characterized ERFs are highlighted with a frame.

(Fig 5B). As revealed by hierarchical clustering of Pearson's correlations, all *DzERF*s for which the expression decreased during ripening were clustered together and were negatively correlated with the ripening-associated genes. However, the *DzERF*s that increased during ripening were clustered together with the ripening-associated genes, suggesting a positive correlation between those *DzERF*s and ripening-related genes (Fig 5A). Notably, as shown in Fig 5B, all *DzERF*s for which the expression increased during ripening exhibited positive correlations

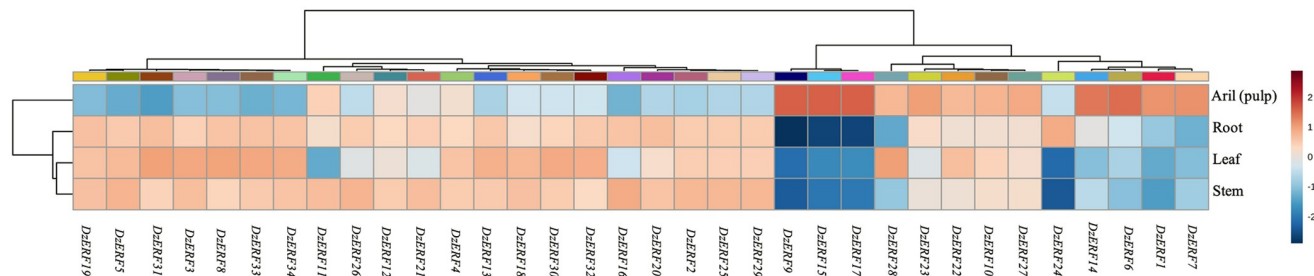

**Fig 4. Tissue-specific expression profile of ripening-associated durian *ERF*s (*DzERF*s) in the Musang King cultivar at the ripe stage.** We used the publicly available Illumina RNA-seq data to analyze the expression levels of ripening-associated *DzERF*s in root, stem, leaf, and fruit pulp tissues. For each *DzERF*, higher expression is presented in red; otherwise, blue is used. The heatmap was generated using MetaboAnalyst 4.0, an open source R-based program. Data were sum-normalized, log transformed, and autoscaled.

with ripening-associated genes. Among these, the highest positive correlation was observed between *DzERF9* and ethylene biosynthetic genes (*SAM synthase*, *ACS*, and *ACO*), followed by *DzERF9* and auxin biosynthetic genes (*TAA1* and *YUCCA4*). However, for those *DzERF*s that decreased during ripening, a negative correlation was observed with ripening-associated genes. Among the DzERFs, the highest negative correlation was found between *DzERF6* and ethylene biosynthetic genes (*SAM synthase*, *ACS*, and *ACO*; Fig 5B). We also included a member of the auxin response factor TF family (DzARF2A) in our correlation network analysis.

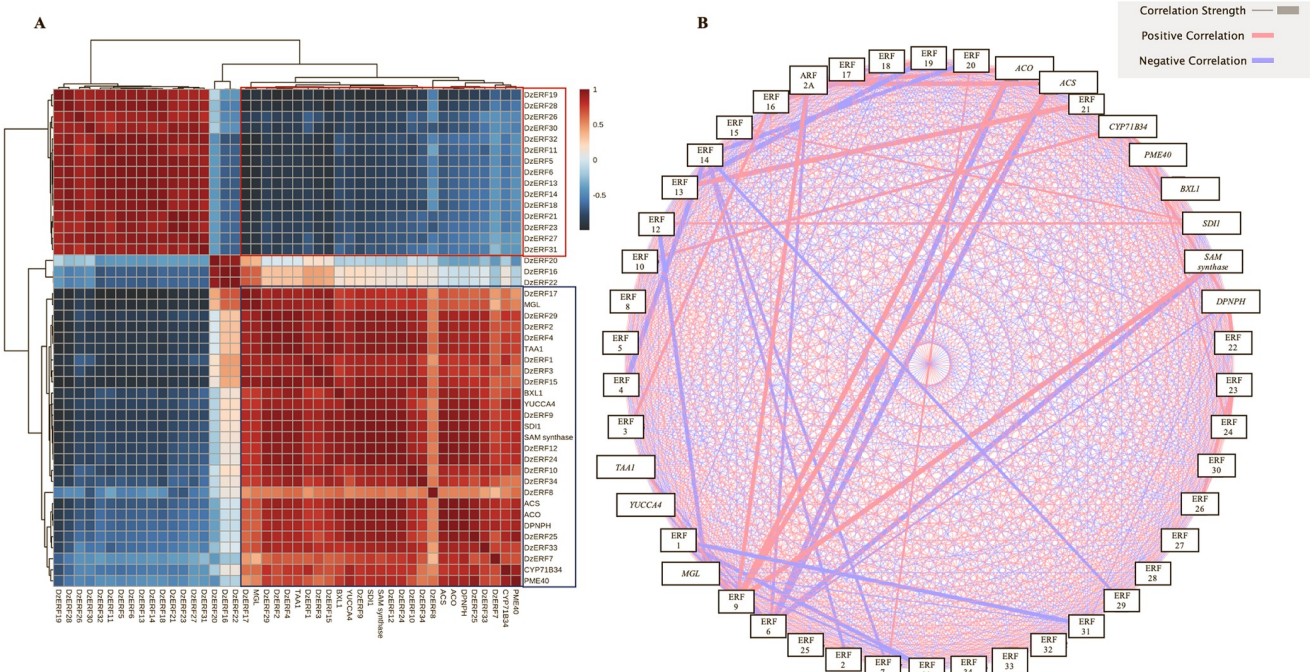

**Fig 5. Gene expression correlation of ripening-associated durian ERFs (DzERFs). (A)** Heatmap of hierarchical clustering of Pearson's correlations (*R*) for 34 ripening-associated DzERFs and previously identified ripening-related genes. Genes with a normalized expression level (RPKM) > 1 were log$_2$ transformed before analysis and were designated as expressed. The DzERFs for which the expression decreased are highlighted with a red frame. The DzERFs for which the expression increased and ripening-related genes are highlighted with a blue frame. **(B)** Correlation network analysis of 34 ripening-associated DzERFs, previously identified ripening-related genes, and a previously characterized member of the ARF TF family (DzARF2A). The thickness of the line corresponds to the correlation strength. Red lines represent positive correlations, whereas blue lines indicate negative correlations.

This previously characterized TF was shown to transactivate ethylene biosynthetic genes (Khaksar and Sirikantaramas, 2020). A positive correlation was observed between *DzERF9* and *DzARF2A*, whereas *DzERF6* was negatively correlated with *DzERF2A* (Fig 5B). Taking into account both the strong correlation with ethylene biosynthetic genes and the pattern of expression during fruit ripening, DzERF6 and DzERF9 were selected as candidates for repressing and activating durian fruit ripening, respectively.

## RT-qPCR analysis

We used RT-qPCR to examine and validate the expression levels of candidate *DzERF6* and *DzERF9* during the post-harvest ripening of durian fruit cv. Monthong. *DzERF6* expression was decreased during ripening (Fig 6A). However, the *DzERF9* expression pattern increased during ripening, with a peak at the ripe stage (Fig 6B). The transcript accumulation patterns of our selected *DzERF*s were consistent with the data obtained through transcriptomics. In addition, we profiled the expression levels of *DzERF6* and *DzERF9* under three different ripening conditions, ethephon-induced, natural, and 1-MCP-delayed ripening. Notably, the expression level of *DzERF6* was significantly repressed under ethephon treatment and was induced by 1-MCP when compared to the control (natural ripening) levels (Fig 6C). However, *DzERF9*

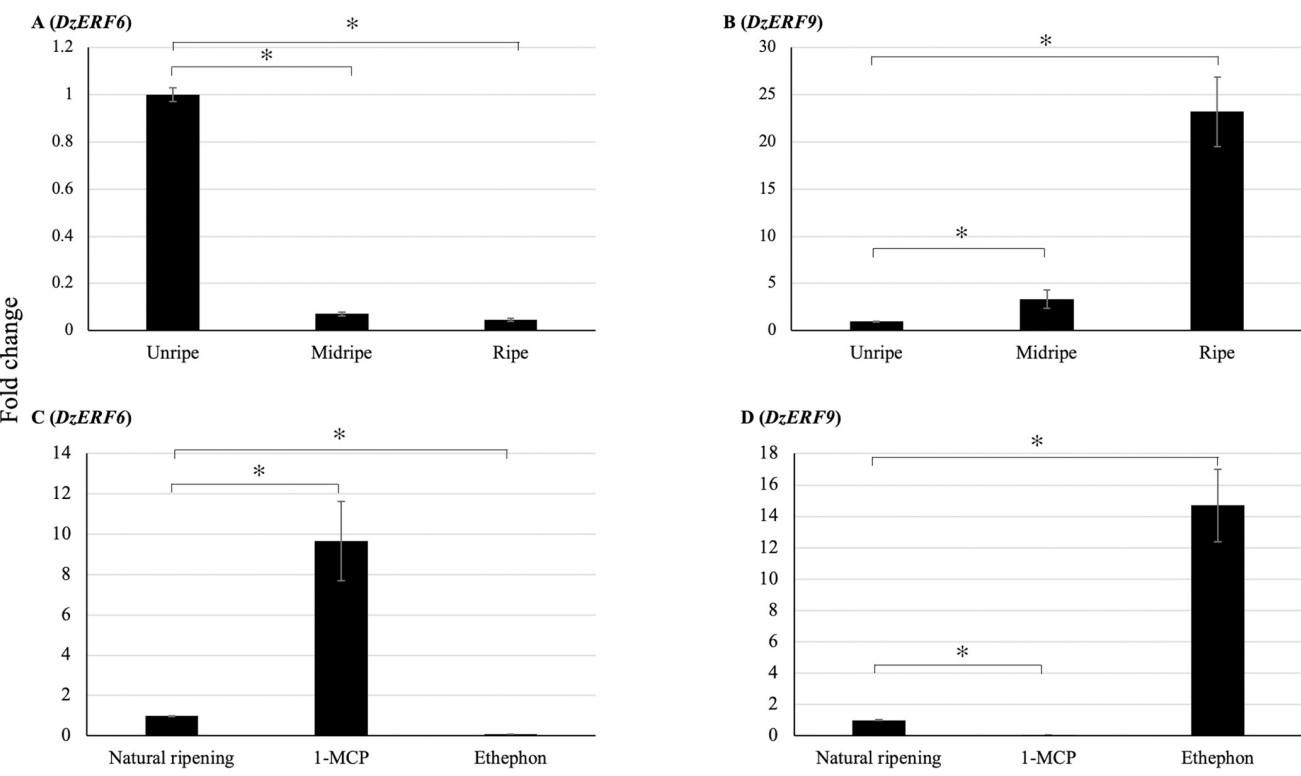

**Fig 6. Fold changes in expression levels of candidate ripening-associated durian *ERF*s (*DzERF*s) at three different stages (unripe, midripe, and ripe) during the post-harvest ripening of durian fruit (Monthong cultivar) and under three different ripening conditions.** (**A** and **B**) The relative gene expression levels of *DzERF6* and *DzERF9* were calculated using the $2^{-\Delta\Delta Ct}$ method, and levels were normalized by the geometric mean of reference genes and the unripe stage as the control. Three independent biological replicates were used. An asterisk above the bars indicates a significant difference at $P < 0.05$ (*). (**C** and **D**) The relative expression levels of *DzERF6* and *DzERF9* were also quantified under three different ripening conditions, natural (control), ethylene-induced, and 1-MCP-delayed ripening by using the $2^{-\Delta\Delta Ct}$ method, and levels were normalized by the geometric mean of reference genes and the natural ripening condition as the control. Three independent biological replicates were used. An asterisk above the bars indicates a significant difference at $P < 0.05$ (*).

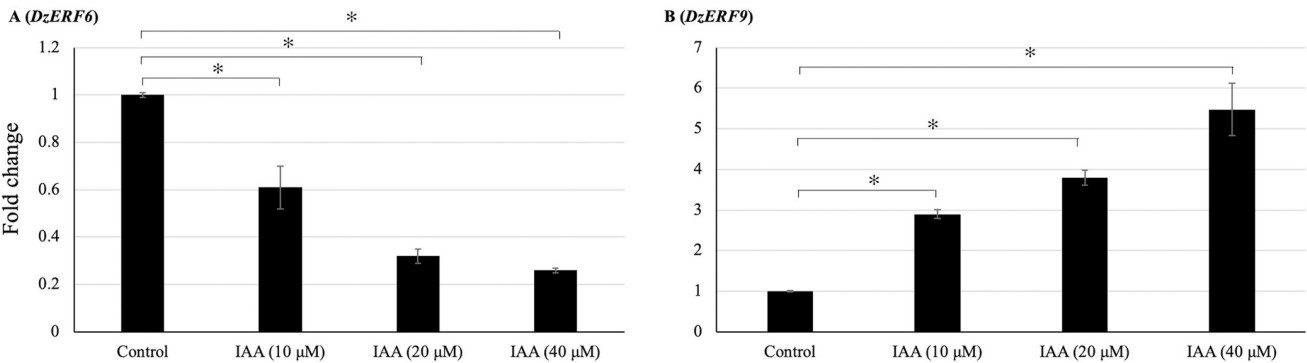

**Fig 7. Auxin-responsiveness of candidate ripening-associated durian *ERF*s (*DzERFs*).** Fold changes in expression levels of *DzERF6* (**A**) and *DzERF9* (**B**) in durian leaves of the Monthong cultivar treated with 0 (control), 10, 20, and 40 μM indole-3-acetic acid (IAA) for 2 h were calculated using the $2^{-\Delta\Delta Ct}$ method, and levels were normalized by the geometric mean of reference genes and the control samples (0 μM IAA). Three independent biological replicates were used. An asterisk above the bars indicates a significant difference at $P < 0.05$ (*).

transcript accumulation significantly increased under ethephon treatment and dramatically decreased with 1-MCP relative to that in the control (Fig 6D). Taken together, our results provide convincing evidence for the role of DzERF6 as a transcriptional repressor and DzERF9 as a transcriptional activator of durian fruit ripening.

## Profiling expression levels of *DzERF6* and *DzERF9* with exogenous auxin treatment

Previously, we found an increasing level of auxin during the post-harvest ripening of durian fruit [32]. Accordingly, we profiled the expression levels of *DzERF6* and *DzERF9* to investigate the auxin-inducibility of their expression patterns. Exogenous auxin treatment significantly repressed the expression level of *DzERF6* in a dose-dependent manner (Fig 7A), whereas for *DzERF9*, we observed significantly higher transcript accumulation with increasing concentrations of auxin (Fig 7B). Exogenous auxin treatment at 40 μM elicited the highest expression level of *DzERF9* (Fig 7B). These results revealed the auxin-responsiveness of both *DzERF6* and *DzERF9* in a concentration-dependent manner and suggested the auxin-mediated role of DzERF6 and DzERF9 in regulating durian fruit ripening.

## Discussion

TFs act as key regulators of gene expression networks that control various developmental and physiological processes in plants, including fruit ripening. The identification and functional characterization of TFs can provide insights for a better understanding of these processes and their associated complex regulatory networks. The ERF TFs comprise one of the largest TF families, which is a part of the AP2/ERF superfamily. The defining characteristic of the members of this superfamily is the highly conserved DBD of approximately 60 amino acids, designated as the AP2/ERF domain. According to the Plant Transcription Factor Database (http://planttfdb.gao-lab.org), 248 members of the *ERF* gene family exist in the durian genome. ERFs act downstream of the ethylene signaling pathway and regulate the expression of ethylene-responsive genes by binding to the conserved motifs in the promoter regions of target genes. It has been well documented that ethylene plays an essential role in initiating and orchestrating climacteric fruit ripening, and ERFs have been assigned as the core of ethylene signaling. Thus, studies on the identification and characterization of ERFs would provide a deeper

understanding of ethylene-dependent ripening. Numerous studies have previously identified the members of the ERF TF family in various crops and documented their key regulatory roles in controlling different aspects of climacteric ripening [20–26]. Nevertheless, little is known about the possible role of ERFs in regulating the expression of ethylene biosynthetic genes in relation to climacteric fruit ripening.

In this study, based on the transcriptome data of durian fruit cv. Monthong at three different stages of post-harvest ripening (unripe, midripe, and ripe), we identified 34 ripening-associated *DzERF*s, designated *DzERF1* to *DzERF34*. Heat map representation according to the expression levels classified *DzERF*s into three separate clades (Fig 1). Clade I consisted of 15 members, with a decreasing expression level during ripening. However, clade III comprised 16 members that were upregulated over the course of ripening (Fig 1).

The domains and motifs of transcription factors are often associated with transcriptional activity, protein-protein interactions, and DNA binding [45]. Conserved motif analyses provided a better understanding of gene evolution and potentially functional differences. A total of 10 motifs were identified, among which motif 1 and 2 contained a wide region of the AP2/ERF domain and were commonly shared among all DzERFs, except for DzERF19, which lacked motif 2 (Fig 2). The functions of other motifs are still unknown and must be further elucidated, as previously stated for ERFs from other species [6, 16, 46]. Although the functions of these motifs have not been investigated, it is plausible that some might play major roles in protein-protein interactions.

Our phylogenetic analysis clustered the 34 ripening-associated DzERFs into 15 subclades, among which some DzERFs were paired with previously characterized ERFs from other fruit crops (Fig 3). Increasing evidence suggests that the identification of characterized orthologues is a powerful tool to predict the functions of genes. Orthologous proteins have similar biological functions in different species [47–49]. Based on our phylogenetic analysis, DzERF6 and DzERF11 were paired with ERF6 of tomato (SlERF6), ERF11 of banana (MaERF11), and ERF2 of apple (MdERF2) in subclade B1 (Fig 3). Therefore, these three ERFs were considered the closest orthologs of DzERF6 and DzERF11. Functional characterization of SlERF6 [21], MaERF11 [24], and MdERF2 [29] suggested their role as transcriptional repressors of fruit ripening that function by targeting the promoter of ethylene biosynthetic genes and negatively regulating their transcription. This finding strengthened the possibility of a similar role for DzERF6 and DzERF11, which were downregulated during durian fruit ripening. In subclade B4, DzERF9 was paired with ERFs from banana (MaERF9), pear (PpERF24), and tomato (SlERFB3) (Fig 3). These three orthologs of DzERF9 were experimentally confirmed to act as positive regulators of fruit ripening via the transcriptional regulation of ethylene biosynthetic genes [22, 28, 36]. These findings, along with the marked increase in expression levels during ripening, indicate the possible role of DzERF9 as a transcriptional activator of ripening via the regulation of climacteric ethylene biosynthesis. Notably, our *in silico* analysis of the promoter regions of *ACS* and *ACO* of durian revealed the existence of binding sites for ERF TFs, specifically the GCC box (AGCCGCC) and/or dehydration-responsive element/C-repeat (DRE/CRT) (CCGAC) (S4 Fig). Consistently, the amino acid sequence analysis of DzERF9 showed regions of acidic amino acid-rich, including Gln-rich and/or Ser/Thr-rich amino acid sequences which are often designated as transcriptional activation domains [50]. However, our sequence analysis of DzERF6 revealed the existence of regions rich in DLN(L/F)xP, which are often associated with transcriptional repression [51].

In addition to the potential role of DzERFs in mediating fruit ripening by regulating climacteric ethylene biosynthesis, our phylogenetic analysis suggested other roles of DzERFs in various aspects of ripening. In subclade D3, DzERF21 was paired with ERFs from papaya (CpERF9) [25], kiwi (AdERF9) [23], peach (ppeERF2) [37], and persimmon (DkERF8/16/19)

[38] (Fig 3). Functional characterization of these ERFs confirmed their roles in ripening via cell wall degradation (fruit softening). Two DzERFs, including DzERF30 and DzERF31, were paired with a member of the ERF from tomato (SlERFPti4) in subclade D4 (Fig 3). SlERFPti4 has been reported to regulate carotenoid biosynthesis during fruit ripening [52]. Taken together, these findings suggest the potential role of DzERFs in regulating various aspects of durian fruit ripening.

To gain a deeper understanding of the roles of DzERFs during fruit ripening, we searched for potential target genes regulated by DzERFs via including the 34 ripening-associated DzERFs through correlation analysis with previously identified ripening-associated genes involved in ethylene biosynthesis, sulfur metabolism, fruit softening, and aroma formation (identified by Teh et al. [31]) and auxin biosynthesis (identified by Khaksar et al. [32]) during durian fruit ripening. All DzERFs that were upregulated during ripening exhibited positive correlations with these genes, with DzERF9 showing the highest positive correlation with *ACS* and *ACO* (Fig 5B). However, the DzERFs that were downregulated during ripening were negatively correlated with the ripening-associated genes, among which DzERF6 had the highest negative correlation with ethylene biosynthetic genes (Fig 5B). These observations, consistent with the roles suggested for DzERF6 and DzERF9 via phylogenetic analysis, implied the potential role of both factors as transcriptional repressors and activators of ripening, respectively, that function via the transcriptional regulation of climacteric ethylene biosynthesis. Accordingly, these two DzERFs were selected as candidate ERFs for further analysis. Notably, we included our previously characterized member of the ARF TF family (DzARF2A) in our correlation network analysis. Consistent with the *in vivo* assay [33], our correlation analysis revealed a positive correlation between DzARF2A and ethylene biosynthetic genes (*ACS* and *ACO*) (Fig 5B). Of particular note, DZARF2A showed a positive correlation with DzERF9, whereas it was negatively correlated with DzERF6 (Fig 5B).

Using RT-qPCR, we profiled the expression levels of our candidate *DzERF*s at three different stages (unripe, midripe, and ripe) during the post-harvest ripening of durian fruit cv. Monthong. The transcript abundance patterns of both *DzERF6* and *DzERF9* were consistent with the data obtained by transcriptomics (Fig 6A and 6B). In addition, we examined the expression levels of *DzERF6* and *DzERF9* under ethephon and 1-MCP treatments. The expression level of *DzERF6* was significantly induced by 1-MCP and suppressed by ethephon (Fig 6C). The ethylene-repressed expression of *DzERF6*, along with the decreasing expression pattern during ripening, strengthened its possible role as a transcriptional repressor of ripening. However, the expression level of *DzERF9* was dramatically upregulated under ethephon treatment and suppressed by 1-MCP (Fig 6D). The ethylene-induced expression of *DzERF9* coincided with an increasing expression level during ripening, suggesting a potential role in regulating the post-harvest ripening of durian fruit as a transcriptional activator. These findings highlighted the marked ripening-associated expression patterns of *DzERF6* and *DzERF9* and prompted us to investigate their regulatory roles during durian fruit ripening.

The idea that climacteric fruit ripening is modulated by a complex hormonal network has already been formulated and suggested in the existing literature. In our previous study, we detected an increasing level of auxin during the post-harvest ripening of durian fruit, suggesting a ripening-associated role of auxin, which has previously been documented for climacteric tomato [53] and peach [54]. Notably, the expression levels of both *DzERF6* and *DzERF9* were found to be responsive to exogenous auxin treatment, but in the opposite manner (Fig 7). Exogenous auxin suppressed the expression level of *DzERF6*. However, the expression level of *DzERF9* was significantly increased with increasing auxin concentrations (Fig 7). This observation strengthened the possibility that DzERF6 and DzERF9 regulate durian fruit ripening in concert with auxin. Notably, our *in silico* analyses of the 2-kb promoter regions located

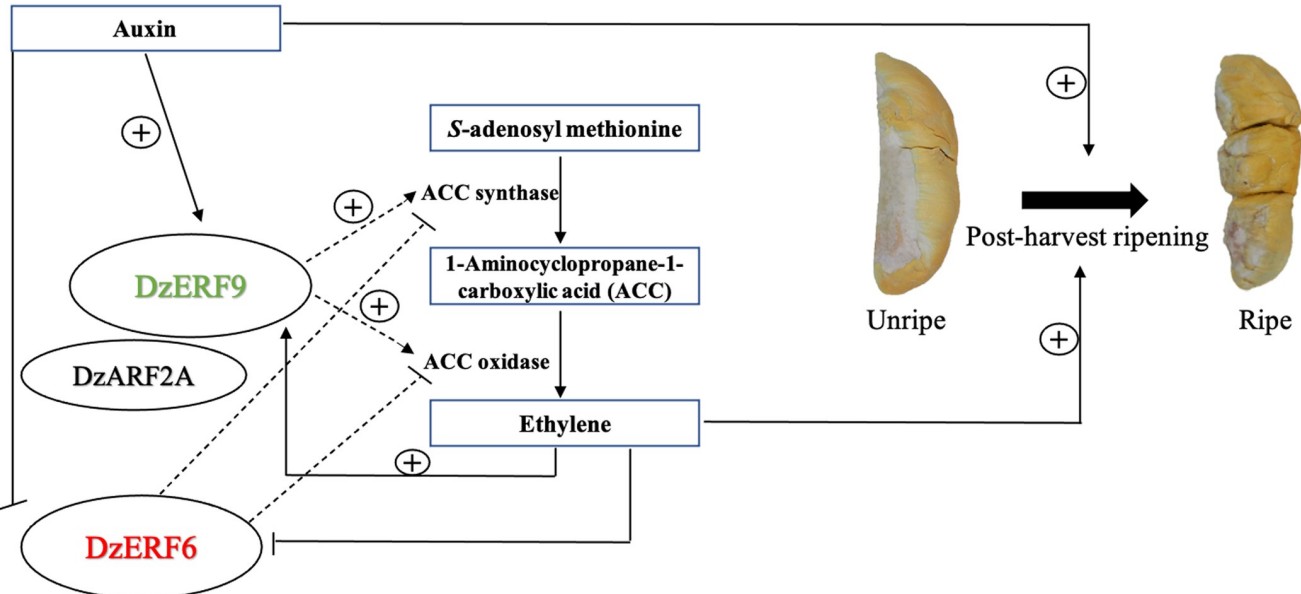

**Fig 8. A general scheme depicting the role of ripening-associated DzERF6 and DzERF9 in the regulatory network mediating durian fruit ripening.** Data from this study and our previous one [33] are integrated and presented. DzERF9 might exert its effect via the ethylene-dependent ripening pathway by positively regulating the transcription of ethylene biosynthetic genes (ACC synthase; *ACS* and ACC oxidase; *ACO*). It appears that the expression of *DzERF9* is positively regulated by both auxin and ethylene. DzARF2A is a positive regulator of fruit ripening that functions by trans-activating the ethylene biosynthetic genes. DzARF2A might interact with DzERF9, and together they act as a positive regulator of durian fruit ripening. As a negative regulator of fruit ripening, DzERF6 represses the transcription of ethylene biosynthetic genes. Arrows indicate positive regulation (activation) whereas blunt-ended lines indicate negative regulation (repression). The dashed lines denote our proposed regulatory role during ripening.

upstream of the translation start site of *DzERF6* and *DzERF9* confirmed the existence of auxin response elements (AuxREs; TGTCTC) which are the binding sites for auxin response factor (ARF) TF family (S5 Fig). In our previous study, we identified a member of the auxin response factor (ARF) TF family, DzARF2A, which mediates durian fruit ripening by trans-activating ethylene biosynthetic genes (Khaksar and Sirikantaramas 2020). Based on our previous findings and the results obtained herein, we propose a regulatory network modulating the post-harvest ripening of durian fruit, which includes not only ERF and ethylene as master regulators but also other TFs and hormones (Fig 8). DzERF9 might function as a transcriptional activator of ripening, activating the expression of master regulators and ethylene biosynthetic genes (*DzACS* and *DzACO*). It is speculated that DzERF9 and DzARF2A obtain signals from auxin and ethylene, both of which induce ethylene biosynthesis. DzARF2A might interact with DzERF9 and/or other TFs to form an enhanceosome and fine-tune durian fruit ripening (Fig 8). As a negative regulator of ripening, the expression level of *DzERF6* was suppressed by auxin and ethylene (Fig 8).

Different TFs can interact to control the expression of a particular gene by forming enhanceosome or repressosome complexes [55]. A few studies have previously documented the interactions among various ripening-associated TFs, including the tomato MADS box FRUITFULL homologs FUL1 and FUL2 interacting with the MADS box protein RIPENING INHIBITOR (RIN) [56], the banana ERF (MaERF9) interacting with MaDof23 [26], and tomato ASR1 (ABA STRESS RIPENING-INDUCED 1) interacting with ARF2A [57]. Investigating the possible interaction between DzERF and other ripening-associated TFs, such as DzARF (as proposed in Fig 8), could be the subject of further study.

In summary, transcriptome-wide identification and expression profiling revealed 34 ripening-associated members of the *ERF* gene family in durian. Among these, the marked ripening-associated expression patterns of *DzERF6* and *DzERF9* and their strong correlation with ethylene biosynthetic genes prompted their further expression profiling under ethylene and auxin treatment conditions. The expression levels of both *DzERF6* and *DzERF9* were responsive to exogenous ethylene and auxin, suggesting a hormonal and transcriptional regulatory network in which ethylene acts in concert with auxin as a master regulator of durian fruit ripening by affecting the expression of ripening-associated *DzERF*s. Our findings provide a deeper understanding of the role of ERF TFs in mediating durian fruit ripening. Further functional characterization of DzERF6 and DzERF9 in fruits would provide more insights into their ripening-associated roles during durian fruit ripening.

## Supporting information

**S1 Table. List of primers for *DzERF*s and reference genes used in this study.**
(PDF)

**S1 Fig. Photos of durian pulp samples.** Representative photos of three types of durian pulp samples (mature (unripe), midripe ($\sim$3 days after harvest), and ripe ($\sim$5 days after harvest)) during post-harvest ripening used in our study.
(PDF)

**S2 Fig. Multiple sequence alignment of the amino acid sequences of the ripening-associated durian ERFs (DzERFs).** Multiple sequence alignment analysis was carried out using ClustalW. A conserved DNA binding domain (DBD) of 61 amino acid residues designated the AP2/ERF domain was found at the N-terminal region of all DzERFs. Identical amino acids are highlighted by color.
(PDF)

**S3 Fig. Multilevel consensus sequences identified by MEME.** Protein sequences of ripening-associated DzERFs were used to identify conserved motifs. Ten conserved motifs were identified. Motifs 1 and 2 represent the conserved DNA binding domain (AP2/ERF domain) always observed at the N-terminus.
(PDF)

**S4 Fig. Nucleotide sequences of the 2-kb promoter regions of ethylene biosynthetic genes from durian (*DzACS* and *DzACO*).** ERF binding sites: GCC box (AGCCGCC) and/or dehydration-responsive element/C-repeat (DRE/CRT) (CCGAC) are highlighted in yellow. The translational start site (ATG) is underlined.
(PDF)

**S5 Fig. Nucleotide sequences of the 2-kb promoter regions of *DzERF6* and *DzERF9*.** Auxin response factor (ARF) binding sites (TGTCTC) are highlighted in yellow. The translational start site (ATG) is underlined.
(PDF)

## Acknowledgments

We thank Kittiya Tantisuwanichkul for assisting in taking durian pulp photos.

## Author Contributions

**Conceptualization:** Supaart Sirikantaramas.

**Formal analysis:** Gholamreza Khaksar, Supaart Sirikantaramas.

**Funding acquisition:** Supaart Sirikantaramas.

**Investigation:** Gholamreza Khaksar.

**Methodology:** Gholamreza Khaksar.

**Supervision:** Supaart Sirikantaramas.

**Visualization:** Gholamreza Khaksar.

**Writing – original draft:** Gholamreza Khaksar.

**Writing – review & editing:** Supaart Sirikantaramas.

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
