## [Decision Letter · Decision Letter 0]

17 Jun 2021

PONE-D-21-15671

Transcriptome-wide identification and expression profiling of the ERF gene family suggest roles as transcriptional activators and repressors of fruit ripening in durian

PLOS ONE

Dear Dr. Sirikantaramas,

Thank you for submitting your manuscript to PLOS ONE. After careful consideration, we feel that it has merit but does not fully meet PLOS ONE’s publication criteria as it currently stands. Therefore, we invite you to submit a revised version of the manuscript that addresses the points raised during the review process.

Your manuscript was reviewed by two scientists with expertise in the subject area of your study.  As you can see from their reviews, both referees positively evaluated your manuscript, but still had several concerns especially on your experimental design including statistical method.  In view of the referees’ reviews and recommendations, the current manuscript could be accepted after major revision.  I hope that the reviews will aid you and colleagues in considering the additional studies. 

From handling editor:

We look forward to receiving your revised manuscript.

Kind regards,

Takaya Moriguchi

Academic Editor

PLOS ONE

Journal Requirements:

Reviewers' comments:

Reviewer's Responses to Questions

**Comments to the Author**

1. Is the manuscript technically sound, and do the data support the conclusions?

Reviewer #1: Yes

Reviewer #2: Yes

2. Has the statistical analysis been performed appropriately and rigorously? 

Reviewer #1: Yes

Reviewer #2: No

3. Have the authors made all data underlying the findings in their manuscript fully available?

Reviewer #1: Yes

Reviewer #2: Yes

4. Is the manuscript presented in an intelligible fashion and written in standard English?

Reviewer #1: Yes

Reviewer #2: Yes

5. Review Comments to the Author

Reviewer #1: In the present study, the ERF gene family was investigated in durian fruit. Two ERF genes (DzEFR6 and DzERF9), which regulated durian fruit ripening, were identified and their regulation on ethylene biosynthesis were discussed. The manuscript was well organized, and the results presented in this study were clear and informative. Thus, I suggested it to be published after some revisions.

1. In the material and method, how the authors determined the maturation of fruit? Why 1 day, 3 days, and 5 days were selected in this study. I think that the maturity indication should be described in the material and method, and the photos of unripe, midripe, and ripe samples should be added.

2. It is suggested to describe the concentrations of ethephon and 1-MCP used in this study. In addition, in Line 137, how many days were the fruit stored until the ethephon-induced samples ripened.

3. In this study, the authors found that DzEFR6 was a transcriptional repressor, while DzERF9 was a transcriptional activator of fruit ripening. What was the sequence difference between the two genes? Were functions of DzEFR6 and DzERF9 investigated in other plants previously?

4. The expression of DzEFR6 and DzERF9 in response to auxin treatment was investigated in leaves. This purpose of this study was to characterize the ERF genes in fruit during the ripening. Thus, it was difficult to understand why leaves were used instead of fruit in the auxin treatment experiment.

Reviewer #2: This work tried to clarify the role of the ERF gene family as transcriptional activators and repressors of fruit ripening in durian. The results and conclusion are reasonable at least for this reviewer.

Please consider the following items to enhance the value of your work.

L134-136

As far as the reviewer reads how to apply ethephon and 1-MCP from your previous study, the related description is not so long. Therefore, how about adding such a description in the current manuscript?

L141-142

It is unclear how to apply auxin.

L235-239, and Figs. 6 & 7

This reviewer is not sure whether we can carry out parametric methods for the data of fold change. In any case, nowadays we should not use Duncan’s multiple range test.

Please confirm and reconsider your statistical analysis. If part of the results is changed, you should revise the descriptions related to them.

6. PLOS authors have the option to publish the peer review history of their article (what does this mean?). If published, this will include your full peer review and any attached files.

Reviewer #1: No

Reviewer #2: No

---

## [Author Response · Author response to Decision Letter 0]

21 Jul 2021

Dear Editor,

We do appreciate your valuable comments. We did try to do our best and revised our manuscript accordingly. 

In Fig. 2, explanation for motifs 3-10 should be added in the legend. 

Response: Many thanks for your valuable comment. The explanation for motifs 3-10 has been added to the Figure 2 legend (Line 284-285) and Discussion (line 406-413). 

2. Resolution for some Figures are quite low, provide good ones. 

Response: Following your comment, all source files have been uploaded with higher resolution (at least 350 dpi). 

3. In Fig.7D, the expression of DzERF9 in the natural ripening fruit should be high, but its expression is low against the expectation. Provide the reason(s) for its low expression level. 

Response: Dear Editor,

According to your comment, we think you mean Fig. 6D. 

In Fig. 6D, the relative expression levels of DzERF9 under three different ripening conditions, natural (control), ethylene-induced, and 1-MCP-delayed ripening were quantified by using the 2−ΔΔCt method, and levels were normalized by the geometric mean of reference genes and the natural ripening condition which was set to 1 (as the control) (Line 365-366).

Considering the Ct value of natural ripening, it was significantly higher than 1-MCP and dramatically lower than ethephon, consistent with our expectation. 

4. Provide the data if the existence of the auxin responsive motifs in the promoter region of DzERF6 and DzERF9, if any. 

Response: Many thanks for raising this interesting point.

Our in silico analyses of the 2-kb promoter regions located upstream of the translation start site of DzERF6 and DzERF9 confirmed the existence of auxin response elements (AuxREs; TGTCTC) which are the binding sites for auxin response factor (ARF) transcription factor family (Line 484-486). The promoter regions have been added to the manuscript as Supplementary Fig. S5. 

5. In the current Fig. 8, the role of DzERF6 was not well shown. DzERF6 acts as a repressor in the unripe fruit. Modify the schematic flow showing the DzERF6 function. 

Response: Many thanks for your valuable comment. As a transcriptional repressor of fruit ripening, we propose that DzERF6 acts via negatively regulating the transcription of ethylene biosynthetic genes. Accordingly, this proposed function has been added to our schematic flow. The figure legend has been revised accordingly (Line 504-506).

Dear Reviewer 1,

We do appreciate your valuable comments. We did try to do our best and revised our manuscript accordingly. 

1. In the material and method, how the authors determined the maturation of fruit? Why 1 day, 3 days, and 5 days were selected in this study. I think that the maturity indication should be described in the material and method, and the photos of unripe, midripe, and ripe samples should be added.

Response: Many thanks for your valuable comment. Since durian is a climacteric fruit, the fruit is commonly harvested at a commercially mature stage (105 days after anthesis for Monthong cultivar as mentioned in the manuscript, Line 122). At this stage, the fruit is considered to be at the unripe stage. After leaving the fruit at room temperature for a few days, durian fruit undergoes a post-harvest ripening process during which climacteric ethylene biosynthesis and respiration rate increase dramatically which then accelerate the post-harvest ripening process (Ketsa and Daengkanit, 1998, link: https://www.tandfonline.com/doi/abs/10.1080/14620316.1998.11511017). Therefore, we do not use “day after anthesis: DAA” during post-harvest ripening process which includes midripe and ripe stages. Preferably, to determine the ripeness of the fruit, we count the days during post-harvest ripening process as “days after harvest” and measure fruit firmness as an indication of fruit ripening. Accordingly, fruits which were harvested at the commercially mature stage were kept at room temperature (30 °C) for post-harvest ripening until reaching a firmness of 3.4 ± 0.81 N (∼3 days after harvest) (for midripe stage) and 1.55 ± 0.45 N (∼5 days after harvest) (for ripe stage) (Khaksar et al., 2019). 

We have revised this section accordingly and added the information of fruit firmness (Line 123-127).

Following your comment, the photos of unripe, midripe, and ripe durian samples have been added as Supplementary Fig. S1 (Line 130-131). 

2. It is suggested to describe the concentrations of ethephon and 1-MCP used in this study. In addition, in Line 137, how many days were the fruit stored until the ethephon-induced samples ripened. 

Response: Following your comment, the concentrations of ethephon and 1-MCP were added (Line 137-141). In addition, the control and treated samples were kept at room temperature (30 °C) (for ∼3 days) until the ethephon-induced samples ripened. This point was added to the manuscript (Line 142). 

3. In this study, the authors found that DzEFR6 was a transcriptional repressor, while DzERF9 was a transcriptional activator of fruit ripening. What was the sequence difference between the two genes? Were functions of DzEFR6 and DzERF9 investigated in other plants previously? 

Response: Many thanks for your valuable comment. Following your suggestion, we analyzed and compared the sequences of DzERF6 and DzERF9. Notably, the amino acid sequence analysis of DzERF9 showed regions of acidic amino acid-rich, including Gln-rich and/or Ser/Thr-rich amino acid sequences which are often designated as transcriptional activation domains [Liu et al., 1999]. However, our sequence analysis of DzERF6 revealed the existence of regions rich in DLN(L/F)xP, which are often associated with transcriptional repression [Ohta et al., 2001]. This point has been added to our discussion (Line 432-436).

The functions of both DzERF6 and DzERF9 have been previously investigated. Functional characterization of ERF6 from tomato (SlERF6) suggested its role as a transcriptional repressor via negatively regulating transcription of ethylene biosynthetic genes (Line 421-423).

Functional characterization of ERF9 from banana (MaERF9) suggested its role as a transcriptional activator of fruit ripening via trans-activating the ethylene biosynthetic genes (Line 425-427).

4. The expression of DzEFR6 and DzERF9 in response to auxin treatment was investigated in leaves. This purpose of this study was to characterize the ERF genes in fruit during the ripening. Thus, it was difficult to understand why leaves were used instead of fruit in the auxin treatment experiment. 

Response: Many thanks for raising this point.

Unfortunately, during conducting this experiment, we did not have durian fruits to be used in our experiment because it was not the fruit season in Thailand. Therefore, we only had access to the leaves of durian trees. We hope you accept our explanation. Although we performed our experiment in durian leaves, the results provided a strong evidence that auxin affects the expression of both DzERF6 and DzERF9. 

More details regarding the auxin treatment have been added to the Materials and methods (Line 145-148). 

Dear Reviewer 2,

We do appreciate your valuable comments. We did try to do our best and revised our manuscript accordingly. 

L134-136

As far as the reviewer reads how to apply ethephon and 1-MCP from your previous study, the related description is not so long. Therefore, how about adding such a description in the current manuscript? 

Response: Many thanks for your valuable comment. Following your suggestion, detailed description has been added to the manuscript (Line 137-144). 

L141-142

It is unclear how to apply auxin. 

Response: Many thanks for your valuable comment. Following your suggestion, detailed information regarding the auxin treatment has been added to the manuscript (Line 145-148). 

L235-239, and Figs. 6 & 7

This reviewer is not sure whether we can carry out parametric methods for the data of fold change. In any case, nowadays we should not use Duncan’s multiple range test.

Please confirm and reconsider your statistical analysis. If part of the results is changed, you should revise the descriptions related to them.

Response: We do appreciate your valuable suggestion. Following your comment, we reconsidered our statistical analysis. We carried out our statistical analysis of RT-qPCR data using the method described in Steibel et al. (2009) which consists of the analysis of cycles to threshold values (Ct) for the target and reference genes according to a linear mixed model (Model I of the report of Steibel et al. (2009)). This approach is more accurate and powerful than existing alternatives (parametric methods) for RT-qPCR data analysis and has been extensively exploited in many studies, such as:

Pathi KM, Rink P, Budhagatapalli N, Betz R, Saado I, Hiekel S, Becker M, Djamei A and Kumlehn J (2020) Engineering Smut Resistance in Maize by Site-Directed Mutagenesis of LIPOXYGENASE 3. Front. Plant Sci. 11:543895. doi: 10.3389/fpls.2020.543895 

Becker, M., Ngo, N.S. & Schenk, M.K.A. Silicon reduces the iron uptake in rice and induces iron homeostasis related genes. Sci Rep 10, 5079 (2020). https://doi.org/10.1038/s41598-020-61718-4

Angrimani DSR, Francischini MCP, Brito MM, Vannucchi CI. Prostatic hyperplasia: Vascularization, hemodynamic and hormonal analysis of dogs treated with finasteride or orchiectomy. PLoS One. 2020 Jun 25;15(6):e0234714. doi: 10.1371/journal.pone.0234714 

Khaksar G and Sirikantaramas S (2020) Auxin Response Factor 2A Is Part of the Regulatory Network Mediating Fruit Ripening Through Auxin-Ethylene Crosstalk in Durian. Front. Plant Sci. 11:543747. doi: 10.3389/fpls.2020.543747

Dimopoulou, A., Theologidis, I., Liebmann, B. et al. Bacillus amyloliquefaciens MBI600 differentially induces tomato defense signaling pathways depending on plant part and dose of application. Sci Rep 9, 19120 (2019). https://doi.org/10.1038/s41598-019-55645-2

Ayuso M, Fernández A, Núñez Y, Benítez R, Isabel B, Barragán C, et al. (2015) Comparative Analysis of Muscle Transcriptome between Pig Genotypes Identifies Genes and Regulatory Mechanisms Associated to Growth, Fatness and Metabolism. PLoS ONE 10(12): e0145162. doi:10.1371/journal.pone.0145162

Pe ´rez-Montarelo D, Ferna ´ndez A, Barraga ´n C, Noguera JL, Folch JM, et al. (2013) Transcriptional Characterization of Porcine Leptin and Leptin Receptor Genes. PLoS ONE 8(6): e66398. doi:10.1371/journal.pone.0066398

Accordingly, we revised the manuscript text as follows: Materials and methods (Line 241-247), Fig. 6 legend (Line 362 & 367), and Fig. 7 legend (Line 381-382).

---

## [Editor Report · Decision Letter 1]

26 Jul 2021

Transcriptome-wide identification and expression profiling of the ERF gene family suggest roles as transcriptional activators and repressors of fruit ripening in durian

PONE-D-21-15671R1

Dear Dr. Sirikantaramas,

We’re pleased to inform you that your manuscript has been judged scientifically suitable for publication and will be formally accepted for publication once it meets all outstanding technical requirements.

Kind regards,

Takaya Moriguchi

Academic Editor

PLOS ONE

Additional Editor Comments (optional):

Dear Dr. Supaart Sirikantaramas, Chulalongkorn University

I have satisfied your revision.
---

## [Editor Report · Acceptance letter]

29 Jul 2021

PONE-D-21-15671R1 

Transcriptome-wide identification and expression profiling of the ERF gene family suggest roles as transcriptional activators and repressors of fruit ripening in durian 

Dear Dr. Sirikantaramas:

I'm pleased to inform you that your manuscript has been deemed suitable for publication in PLOS ONE. Congratulations! Your manuscript is now with our production department. 

Kind regards, 

on behalf of

Dr. Takaya Moriguchi 

Academic Editor

PLOS ONE